# Translational Medicine in Acute Ischemic Stroke and Traumatic Brain Injury—NeuroAiD Trials, from Traditional Beliefs to Evidence-Based Therapy

**DOI:** 10.3390/biom14060680

**Published:** 2024-06-11

**Authors:** Narayanaswamy Venketasubramanian, Tseng Tsai Yeo, Christopher Li Hsian Chen

**Affiliations:** 1Raffles Neuroscience Centre, Raffles Hospital, 585 North Bridge Road, Singapore 188770, Singapore; 2Division of Neurosurgery, Department of Surgery, National University Hospital, 5 Lower Kent Ridge Road, Singapore 119074, Singapore; drttyeo@yahoo.com; 3Memory Aging and Cognition Centre, Department of Pharmacology, Yong Loo Lin School of Medicine, National University of Singapore, Blk MD3, 16 Medical Drive, #04-01, Singapore 117600, Singapore; cplhchen@yahoo.com.sg

**Keywords:** MLC901, MLC601, NeuroAiD, stroke, traumatic brain injury, TBI, pathophysiology, neurorepair, neurorestoration, clinical studies

## Abstract

Acute ischemic stroke (AIS) and traumatic brain injury (TBI) are two severe neurological events, both being major causes of death and prolonged impairment. Their incidence continues to rise due to the global increase in the number of people at risk, representing a significant burden on those remaining impaired, their families, and society. These molecular and cellular mechanisms of both stroke and TBI present similarities that can be targeted by treatments with a multimodal mode of action, such as traditional Chinese medicine. Therefore, we performed a detailed review of the preclinical and clinical development of MLC901 (NeuroAiD^TM^II), a natural multi-herbal formulation targeting several biological pathways at the origin of the clinical deficits. The endogenous neurobiological processes of self-repair initiated by the brain in response to the onset of brain injury are often insufficient to achieve complete recovery of impaired functions. This review of MLC901 and its parent formulation MLC601 confirms that it amplifies the natural self-repair process of brain tissue after AIS or TBI. Following AIS and TBI where "time is brain", many patients enter the post-acute phase with their functions still impaired, a period when "the brain needs time to repair itself". The treatment goal must be to accelerate recovery as much as possible. MLC901/601 demonstrated a significant reduction by 18 months of recovery time compared to a placebo, indicating strong potential for facilitating the improvement of health outcomes and the more efficient use of healthcare resources.

## 1. Introduction

Acute ischemic stroke (AIS) and traumatic brain injury (TBI) are two severe neurological events that are both major causes of death and prolonged impairment [1,2]. Despite widely disseminated preventative recommendations to the general population, particularly in high-income countries (HIC), their incidence continues to rise due to the global increase in people exposed to their risk of onset (i.e., the elderly population with at-risk lifestyles increases the incidence of AIS and the number of traumatic events for TBI) [3]. This is especially true in low- and middle-income countries (LMICs) and the poorer parts of HICs.

Overall, due to a lack of effective and safe treatment to improve recovery beyond acute targets (revascularization and organized care for AIS, control of intracranial pressure for TBI) [4], AIS and TBI, due to the complex and costly medical care they necessitate, represent a huge burden for people who remain impaired and for their families and society as well [5,6]. In addition, both stroke and TBI can be responsible for the onset of cognitive disorders and even dementia in the longer term [7,8].

The purpose of this review article is to assess the differences and, more importantly, the similarities in the consequences of stroke and TBI, particularly those associated with the multiple pathophysiological processes activated by these two types of cerebral accidents [9]. As part of a review of the pathophysiological mechanisms involved in stroke and TBI, we will examine those targeted by a post-acute intervention with a natural product, NeuroAiD (MLC901/601). Developed over many years of research from traditional medicine to evidence-based therapy, it possesses a multitarget mode of action with preclinical and clinical results obtained in both stroke and TBI as we will summarize in this article. 

## 2. Materials and Methods

This review focuses on stroke and TBI through a search of English reviews and original articles in PubMed, Clintrials.gov, Z-library, and Google with the following keywords: stroke, traumatic brain injuries, TBI, impairment, dementia, Alzheimer’s, new drugs, pharmacology, clinical development, NeuroAiD, MLC601, MLC901, traditional medicines, and TCM. This primary search was expanded by looking through the bibliographies of systematic review articles. By using the keywords included in the titles of every paragraph of this article, we selected the most relevant and recent articles for stroke, TBI, and dementia for this review. 

According to the PRISMA Checklist 2020 (http://prisma-statement.org/PRISMAStatement/Checklist, accessed on 8 March 2024), we used the following selection criteria for our literature search about stroke and traumatic brain injury.
Inclusion criteria: animal studies; clinical studies in subjects or patients with stroke or brain lesion/injury or cerebral lesion/injury or traumatic brain injury or TBI; placebo-controlled experimental preclinical and clinical studies; observational and epidemiological studies; most recent publications since 2009 apart from less recent studies quoted as reference papers in review publications.Exclusion criteria: publications in a language other than English.
For the search about NeuroAiD, MLC601, MLC901, and their ingredients, we included all of the publications on preclinical or clinical studies containing NeuroAiD or MLC601 or MLC901 or Danqi Piantang Jiaonang and all of the inclusion criteria listed above.As a result, 108 references were chosen for this review after screening more than 200 articles for stroke and TBI reviews in addition to articles about MLC901/601.



## 3. Acute Ischemic Stroke (AIS) and Traumatic Brain Injury (TBI)

AIS and TBI are two types of brain insults leading to similar alterations, involving common brain self-repair mechanisms. They are part of the group of acquired brain injury (ABI), which is divided into two main categories, namely (i) TBI or (ii) non-TBI, to which stroke belongs (Figure 1) [10]. Although not belonging to the same categories of ABI due to certain differences, AIS and TBI nevertheless have essential points in common. As for their differences, they mainly concern their etiology and epidemiology, while their similarities mainly concern their pathophysiology (Figure 1) [10].

### 3.1. Epidemiology

ABI, encompassing both AIS and TBI, has been reported as one of the leading causes of disability and mortality in adults [11,12]. While TBI poses a significant threat to the lives and well-being of children and young adults, the incidence of AIS doubles with each passing decade among individuals aged over 55. Notably, men and women have a 1:20 risk of suffering a stroke before age 70. The prevalence of AIS is higher in men than women, but as individuals age, women tend to have a higher incidence than men.

The worldwide incidences of AIS and TBI per 100,000 are 203 (189 to 218) and 369 (331 to 412), respectively, leading to a significant number of neurological disabilities affecting activities of daily living (ADL) [1]. The GBD 2016 Neurology Collaborators’ study reported 13.7 million new AIS cases, with a total of 80.1 million prevalent cases worldwide. Among these, 41.1 million cases were in women and 39.0 million were in men, with a total of 5.5 million deaths [5]. From 1990 to 2016, the average worldwide lifetime risk of AIS increased from 22.8% to 24.9%, a relative increase of 8.9%. The regions with the highest rates were East Asia and Central and Eastern Europe [13].

Concerning TBI, one important consideration is that a great majority of cases (~90%) are mild TBI, which usually does not result in death but eventually in mood disorders, cognitive deficits, or even dementia in the longer term. In 2016, there were about 27 million new TBI cases and 56 million prevalent cases documented [6]. However, TBIs may be underestimated because injury monitoring or reporting systems do not exist in many parts of the world, particularly in LMICs [14]. The incidence of TBI is highest in HIC, particularly in North America and Europe, where more reliable data are available [15]. It is worth mentioning that while TBI can lead to death, the GBD 2016 TBI report does not classify it as the direct cause of death. Instead, the cause of death is attributed to the initial event resulting in TBI (e.g., falls, road accidents, etc.). However, a systematic review and meta-analysis of TBI found that there was significant heterogeneity among the studies, with a limited number of studies eligible for meta-analysis [16].

### 3.2. Pathophysiology

Overall, in terms of pathogenesis, there are generally more similarities than differences between cerebral ischemia and trauma.

#### 3.2.1. Initial Phase

Starting with differences, AIS and TBI injuries result from very different initial insults [9,17]. AIS is often referred to as an “internal brain trauma” that arises from a transient or permanent blockage in a cerebral artery, restricting the blood flow within the cerebral region supplied by this artery and leading to a poorly perfused penumbra. The main causes of AIS are typically embolic or thrombotic events or cerebral small vessel disease. TBI is primarily caused by an “external brain trauma” that triggers an initial acceleration or deceleration of the brain within the skull. This mechanical impact leads to cerebral damage, disruption of the blood–brain barrier (BBB), deterioration of the vascular system, and brain tissue injuries. Among the multiple causes of TBI, the main ones are motor vehicle accidents, falls, sports injuries, physical assaults, explosions, and military combat [18,19]. The pathophysiological cascades triggered after the occurrence of AIS or TBI show a certain degree of similarity in the molecular and cellular mechanisms leading to cell death and contributing to the development of injury-related disorders (Figure 2) [9,17].

#### 3.2.2. Acute Stage

At the acute stage, both AIS and TBI insults result in a well-known triad of synaptic neurodegeneration involving excitotoxicity, calcium, and mitochondria [20]. Perfusion and metabolic dysregulation can induce prolonged hypoglycemia, resulting in glucose delivery reduction and rapid depletion of adenosine triphosphate (ATP). Ultimately, this leads to the apoptosis and necrosis of grafted cells, like ischemic stroke [20]. The regulation of ionic gradients across cell membranes by ATP-dependent ion pumps is disrupted, causing abnormal neurotransmitter release and alterations in ion levels such as calcium. This may cause excitotoxicity with neuronal insult, which could damage cells and activate downstream molecules like proteases. While some strategies can improve cell viability, the role of glucose and its metabolites has been largely overlooked as a potential solution. 

Another pathway to consider is that of non-excitatory amino acids (L-alanine, glycine, L-histidine, L-glutamine, L-serine, L-threonine, and taurine), which trigger cellular swelling under hypoxic conditions by increasing levels of extracellular glutamate, the principal excitatory neurotransmitter [21]. Thus, these non-excitatory amino acids (AAs) could promote harmful effects by activating N-methyl-D-aspartate (NMDA) receptors (NMDAR). However, regarding the inefficacy of NMDAR blockade observed in clinical trials with AIS patients, an alternative to reduce ischemic damage might be targeting the AA transporters involved in this process [21].

DAMPs, also known as damage-associated molecular patterns, are endogenous chemicals produced as a danger signal by cells when they are injured or dying. They interact with pattern recognition receptors (PRRs) in the innate immune system, present from birth and throughout life, and they lead to the activation of the first immune response to a harmful foreign substance [22]. While DAMPs contribute to the host’s defense, they can also promote pathological inflammatory responses by triggering inflammasome formation and the activation of astroglial, microglial, neuronal, and endothelial cells. This cellular activation induces the release of both pro- and anti-inflammatory cytokines. The initial brain insult is followed by BBB disruption, allowing peripheral immune cells to infiltrate with the release of cytokines and free radicals, and activation of the complement cascade, intensifying the inflammatory response [23]. Once the blockage of blood flow in AIS is cleared and blood flow returns to normal, there is a risk of oxygen-free radical production and cellular damage during reperfusion.

Neuroinflammation is a crucial factor in the pathophysiology of AIS, starting from the initial endothelial activation following an injury to the neurorepair phases that occur over a period of days to months [24,25]. Cytokine signaling plays a significant role in the pathophysiology of AIS and the subsequent neurorepair process. Similarly, in TBI pathophysiology, the distribution of pro- and anti-inflammatory cytokines in secondary lesions changes over time, impacting the development of post-TBI acute, sub-acute, and chronic disability and recovery [26].

#### 3.2.3. Sub-Acute and Chronic Stages

In the subacute stage of both AIS and TBI, extensive research has focused on neuroprotective strategies targeting the reduction in secondary tissue loss and/or improving functional outcomes [27]. Numerous drugs have demonstrated favorable results in animal models with various types of brain insults, but the translation of these potential interventions into controlled studies in humans has generally failed. 

Indeed, experimental models of AIS and TBI have demonstrated the importance of therapeutic strategies aimed at enhancing the various mechanisms that contribute to the functional recovery of the brain. These strategies should rely on therapeutic agents with multiple effects, combining the ability to prevent the ischemic cascade from expanding from the core to the penumbra, as well as stimulating neurorepair processes by producing new neurons from the subgranular zone (SGZ) of the hippocampus dentate gyrus and recreating functional neuronal networks [28,29]. After injury to brain tissue, cerebral ischemia stimulates neurogenesis in defense response to induced damage [30], and proliferation and migration of precursor cells occur within the germinal niches, leading them to reach the injured site [31]. The recovery of patients after ischemia-induced neuronal damage is facilitated by neurogenesis [32]. This has been demonstrated in studies with transgenic ablation of neural precursor cells where the impairment of germinal niches affects the recovery process [33]. Conversely, the transplantation of neural precursor cells showed that these cells are critical to improving both the post-ischemic neurological recovery process and neuronal survival [34]. Near the damaged area, the expression of angiogenic factors, such as vascular endothelial growth factor (VEGF), leads to the formation of new blood vessels that irrigate the lesion area [35]. This angiogenic process near the stroke site triggers new blood vessel formation, neuronal migration, and neuroblast differentiation via the secretion of growth factors, such as brain-derived neurotrophic factor (BDNF) and VEGF, along with metalloproteinases 2 and 9, which are crucial for tissue remodeling in the wound healing process [36]. Overall, this interaction between endothelial cells and neuroblasts plays a key role in post-ischemic recovery.

### 3.3. Main Therapeutic Approaches for Post-Stroke and Post-TBI Recovery

In this section, we will review the main interventions used in the acute and post-acute stages of AIS and TBI.

#### 3.3.1. Acute Stage Interventions

As the primary insult differs between AIS and TBI, the priority in the acute stage will be revascularization for AIS and the control of cerebral edema for TBI. As it is not possible to reverse the initial insult that leads to TBI [37], the adage “Time is brain” has been used to describe the relationship between time and AIS, emphasizing the extent of brain damage depending on the duration of ischemia [38].

The therapeutic area that has progressed more in recent years is that of AIS. In the first post-stroke hours, two revascularization interventions are recommended, i.e., intravenous thrombolysis (IVT) with tissue plasminogen activator (t-PA) [39] and endovascular thrombectomy (EVT) [40,41,42,43,44]. IVT is recommended up to 4.5 h after AIS onset in cases or in centers where EVT is not practiced, as success rates decrease over time [45]. EVT has emerged as the gold standard for patients with AIS due to intracranial LVO in the anterior circulation [40,41,42,43,44]. However, only 12% of patients are eligible for it due to its short therapeutic window, contraindications, and potential damage from reperfusion/ischemic injury [46]. A recent systematic review and meta-analysis found no differences between EVT alone and its combination with IVT, suggesting the need for further randomized clinical trials [47]. Despite high recanalization rates, functional recovery rates remain low, with only 10% of patients achieving full recovery at 3 months [48]. In a European survey conducted in 44 countries, only 7.3% of incident AIS patients received IVT and 1.9% EVT, with the highest country rates for both types of intervention being 20.6% and 5.6%, respectively [49].

Fortunately, TBIs are most often mild in severity (mTBI), but their diagnosis can nevertheless pose some difficulties due to the subtle and transient nature of the acute symptomatology of altered mental status, as well as the poor sensitivity of diagnostic tests such as CT scans [50]. Furthermore, assessment of the patient several days or weeks after a traumatic event could introduce elements that look like symptoms of mTBI. Since the initial damage causing a TBI cannot be reversed, the goal of care approaches should be to prevent further injuries by keeping cerebral perfusion pressure (CPP) at an adequate level, maintain cerebral blood flow (CBF), and avoid hypotension and hypoxia [37]. Two options to maintain CPP are either to increase mean arterial pressure or to reduce intracranial pressure (ICP). The main goal is to sustain normovolemia and avoid hypotension. ICP reduction may be achieved with simple bedside maneuvers such as head-of-the-bed elevation, hyperosmolar therapy, cerebrospinal fluid drainage, pentobarbital coma, and ultimately decompressive craniectomy [37]. Large lesions may require surgical evacuation depending on their size, examination findings, and ICP measurements. The acute management of TBI also requires one to address the prevention of venous thromboembolism, stress ulcers, and seizures, as well as focusing on optimizing nutrition and metabolism.

Neuroprotection encompasses the therapeutic approaches and mechanisms able to protect the central nervous system (CNS) against tissue damage secondary to acute brain injuries such as AIS or TBI [51]. Shortly after the initial chemical and biological storm unleashed in the first hours of AIS and TBI, the corresponding brain damage spontaneously induces neurorepair processes related to brain neuroplasticity, involving the key mechanisms of neurogenesis, synaptogenesis, and angiogenesis [52]. However, depending on the severity, extent, and location of tissue damage, spontaneous recovery is often incomplete and levels of recovery of neurological functions vary widely. As cited by Toth et al. 2021, in terms of therapeutic strategies, the concept of “neuroprotection may be categorized into three groups: (i) pharmacological intervention with antioxidants, neurotransmitter agonists and antagonists, anti-inflammatory drugs, and natural products, (ii) non-pharmacological intervention with exercise, diet, and acupuncture, and (iii) cellular and genetic approaches with growth/trophic factors” [51]. Initiated at the subacute stage of AIS or TBI in the first 24 h after event onset, neuroprotective pharmacological interventions aim to reduce secondary tissue loss and/or improve functional outcomes. They have been the subject of the most intensive research in both AIS and TBI [26]. Various medications have demonstrated favorable outcomes in animal models of cerebral injuries. Nevertheless, the application of these promising treatments in controlled trials has generally produced disappointing outcomes in humans [26]. 

#### 3.3.2. Post-Acute Brain Injury Rehabilitation

Any type of brain injury most often requires specialized long-term care so that patients can heal and recover their functional independence. The ideal rehabilitation care is delivered by a multidisciplinary team of medical and other healthcare professionals.

Depending on the nature and violence of the initial insult, TBIs can include brain contusions, concussions, hematomas, blood clots, hemorrhage, and cerebral edema [10]. Thus, these forms of damage can have a substantial influence on the body’s function and cause long-term repercussions such as dizziness, headaches, depression, and cognitive deficits [53]. More severe injuries can result in physical, intellectual, behavioral, emotional, and sensory difficulties and communication issues [54]. In such cases, the rehabilitation program will frequently require long-term care with experts in brain trauma damage recovery.

Among the most impactful consequences of AIS, motor function impairments are the most frequent (upper limb weakness, 77.4%) [55]. But many other functions can be impaired, such as urinary continence (48.2%), consciousness (44.7%), swallowing (44.7%), and cognition (43.9%). As in post-TBI rehabilitation, the intervention of a multidisciplinary team will be needed. Recovery time after an AIS will vary depending on the severity of symptoms, with severe cases requiring a comprehensive long-term strategy including physical, speech, and occupational therapies.

In the post-acute stage, when revascularization and/or neuroprotection interventions have failed or cannot be carried out, there may be a role for neurorepair with neurorestorative therapies, which could amplify endogenous brain mechanisms without compromising acute interventions or neuroprotective agents [56]. As an exemplar, we shall review the pharmacological and clinical data on the use of NeuroAiD (MLC901/601) in AIS and TBI, focusing on the translation of preclinical results to clinical benefits.

## 4. MLC901 Development in AIS and TBI

MLC901 (NeuroAiD II) is a natural formulation containing extracts from nine herbal ingredients (0.800 g/caps of Radix astragali, and 0.160 g/caps of each of the following ingredients: *Radix salviae mitorrhizae, Radix paeoniae rubra, Rhizoma chuanxiong, Radix angelicae sinensis, Carthamus tinctorius, Prunus persica, Radix polygalae, and Rhizoma acori tatarinowii*) [57]. It is a simplified formulation of its parent formulation *Danqi Piantang Jiaonang* (MLC601-NeuroAiD), which was approved as a traditional Chinese medicine (TCM) in post-AIS recovery by the China Food and Drug Administration in 2001 [58]. Thus, we will review the pharmacological properties and multimodal actions of MLC901 (Table 1) and their clinical translation supported by the results of clinical studies on recovery after AIS and TBI (see Section 4.2. Clinical translation).

### 4.1. Preclinical Development 

Pharmacological studies on animal and cellular models of ischemic and traumatic brain injuries have demonstrated the neuroprotective and neuroregenerative properties of MLC901 [57] (Figure 3).

#### 4.1.1. Neuronal Network Plasticity and Neurogenesis

Effective stimulation of neurorepair processes (neuroplasticity and neurogenesis) is essential for recovering from brain tissue damage and improving the restoration of brain function [59] (Figure 3). The brain reorganizes cortical maps to some extent after ischemia by utilizing its full range of neuronal plasticity [60]. 

Aligned with findings on the efficacy of MLC901/601 in humans, studies in rodents have demonstrated that MLC901 can prevent the death of compromised neuronal tissues in the penumbra area and improve cognitive deficits and functional outcomes by repairing neuronal circuits [61,62]. In the well-established mouse model of focal ischemia induced by temporary middle cerebral artery occlusion (MCAO) for 60 min, it was observed that both pre- and post-treatment with MLC901 resulted in improved blood flow rates, survival, and neurological recovery by reducing neurodegeneration, without affecting physiological parameters [61]. Administered intraperitoneally immediately after ischemia, MLC901 led to a high survival rate and a strong reduction in the volume of acute ischemic lesions (~50%) compared to ischemic control mice 30 h after ischemia. 

Additionally, MLC901 demonstrated efficacy in prophylaxis. A 6-week oral pretreatment with MLC901 before the induction of ischemia significantly reduced the mortality rate and cerebral infarctions. A rat model of global ischemia induced by a 20-min occlusion of all four major vessels [63] mimics sudden cardiac arrest and subsequent cardiopulmonary resuscitation in humans. As post-treatment administered intraperitoneally for seven days, MLC901 significantly attenuated delayed necrotic and apoptotic neuron death in the vulnerable hippocampal CA1 field after one week of reperfusion [62]. Notably, this protective effect against focal and global ischemia in rodents is still effective when MLC901 is administered as late as 3 h after ischemia onset [57,62].

Treated with MLC901, cortical neurons showed a higher density of neurites with more elongated branches, leading to greater expression of the growth-associated protein (GAP) 43 in neurites [61]. This neuroregenerative effect of MLC901 is associated with increased synaptogenesis [61], as seen by the higher expression of synaptotagmin-1, an important synaptic vesicle for synapse formation and function [64,65]. 

These findings indicate that MLC901 may enhance the brain’s natural ability to recover after ischemia by promoting neurogenesis, neurite outgrowth, and synaptogenesis. As previously described, the MLC901 mode of action could involve its ability to enhance the release of BDNF, a critical growth factor regulating neuronal survival [66] and neurogenesis and brain plasticity [67]. In vitro studies indicated that 6-week treatment with MLC901 increased BDNF expression in cortical neurons [61].

#### 4.1.2. Angiogenesis and Neovascularization

In the chronic phase following an AIS, pharmacological treatments targeting the promotion of brain angiogenesis showed promising effects on the long-term recovery of AIS patients [68]. The formation of new blood vessels plays a crucial role in increasing collateral circulation and supporting the survival of new neurons in the brain. This occurs 3 or 4 days after an AIS in humans [69]. The process of neurogenesis, involving the generation of new neurons, is dependent on adequate blood supply for the survival of these newly formed cells. [70]. Studies in rodents have shown that genes involved in angiogenesis are activated shortly after AIS onset, with elevated levels of angiogenic proteins persisting in the affected area for an extended period [71]. Therapeutic angiogenesis is considered a regenerative mechanism that could enhance patient post-stroke outcomes. Of particular interest in modulating angiogenesis and vascular remodeling are factors such as VEGF and the angiopoietin/Tie receptor system [72,73,74].

MLC901 enhances the growth of endothelial cell proliferation and the formation of new blood vessels in the infarct region by increasing the number of neocortical vessels [75]. The activity of MLC901 in regulating hypoxic inducible factor 1α (HIF 1α) and its associated molecules, such as VEGF and angiopoietin 1 and 2, contributes to this process. Additionally, this study demonstrates the role of erythropoietin (EPO) plays in MLC901’s stimulation of angiogenesis [75]. 

These findings show that MLC901 has pharmacologic properties promoting the revascularization and repair of neurovascular units after a brain lesion (Figure 4).

#### 4.1.3. Neuroinflammation 

In the development field of new AIS therapies, neuroinflammation is an important biological reaction to cell and tissue injuries, especially in the brain [76]. It is widespread in neurological disorders such as AIS and TBI [77]. The kind and extent of inflammation are determined by the setting, duration, and progression of the original injury (Figure 4). At early phase of an ischemic stroke, the activation of microglia induces release of inflammatory proteins (IL-1, IL-6, TNF α and IL-10), chemokines, reactive oxygen species (ROS) proteases, and prostaglandins [78]. Linked to elevated expression of glial fibrillary acidic protein, reactive astrocytes proliferate and migrate throughout the damaged area, releasing matrix metalloproteinases (MMPs) [79,80]. These MMPs cause damage to the extracellular matrix and promote BBB breakdown [80]. This disruption allows monocytes, T cells, dendritic cells, and circulating neutrophils to penetrate the brain and intensify inflammation [81]. These cells coordinate the inflammatory response following AIS, communicating with each other and ischemic neurons via signaling molecules and cytokines. The role of inflammation in AIS can be both harmful by contributing to infarct expansion early after AIS onset and beneficial by later playing a crucial role in infarct resolution and influencing remodeling and repair processes [82]. Throughout the process of neuroinflammation resolution, tissue regeneration is stimulated by anti-inflammatory cytokines such as IL 10, and trophic factors stimulate tissue regeneration [83]. However, if the inflammatory process is not treated, it might result in persistent CNS inflammation and neurodegeneration.

The favorable effects of MLC901/601 on inflammatory processes have been demonstrated both in animal models of AIS [84] and TBI [85] where microglia activation plays a major role [84]. In a study on a focal ischemia model of AIS with MCAO, Widmann et al. demonstrated that, in addition to its beneficial effects against microglial activation, MLC901 has several other anti-inflammatory properties that affect neuronal cells, the vascular endothelium, and neutrophil infiltration [84]. The authors assume that these numerous beneficial effects of MLC901 on inflammation and neurorepair are likely the result of its composition of multiple herbal ingredients. In a TBI-induced brain contusion in rats, the effects of early therapy with MLC601 after TBI (1 h post TBI) were shown to be significant in a dose-dependent manner [85]. The reduction in microglial activation resulted in a reduction in neuronal death and neurological and motor impairments. A previous study also showed that MLC901 therapy had neuroprotective and neuroregenerative benefits after TBI in rats [86]. These results support the potential of MLC901 to improve neurorepair with neurological and motor functional recovery both in AIS and TBI patients. 

### 4.2. Clinical Translation

Concerning neuroprotection, the complexity of the pathophysiology of brain lesions due to cerebral ischemia or trauma is a major hurdle to the successful translation of preclinical findings into the effective clinical treatment of AIS [26]. To address this challenge, research has been undertaken to identify agents that can disrupt multiple signaling pathways or develop treatment combinations able to sequentially stop ischemic injury, particularly in the penumbra or the region surrounding the core lesion. However, despite promising results in animal models of AIS, none of the neuroprotective agents tested have achieved significant clinical results [46]. 

Beyond the acute phase, which is the time of revascularization attempts for AIS and intensive care for TBI, comes the time of neurovascular repair mechanisms which can be summarized in four main processes: (i) neuronal network plasticity, (ii) neurogenesis, (iii) angiogenesis, and (iv) anti-inflammation with gliosis [87]. As shown in the preclinical models (Figure 3), MLC901 enhances these neurovascular repair mechanisms with its multimodal actions of up- or down-regulation on growth factors (BDNF, VEGF, and angiopoietins 1 and 2), cytokines (TNFα and interleukins), chemokines (CCL2), and other mediators (HIFα, TLR-4, and Prx-6). The objective of the next section is to review how these multi-herbal formulation properties have translated to the clinical level as a support for the brain’s self-repair process.

#### 4.2.1. Clinical Studies on AIS

The clinical development program of MLC901/601 was first initiated in the management of AIS recovery and oriented to the neurorepair phase by starting the treatment at the post-acute and chronic stages of AIS. 

##### TCM Development in China

Clinical development started with Danqi Piantang Jiaonang (DPJ), the TCM name for MLC601, indicated in China to improve recovery after AIS. DPJ was compared to another TCM, Buchang Naoxintong Jiaonang (BNJ), in two early double-blind, randomized clinical trials [58]. Chen et al. 2000 reported the outcomes of these studies showing the effectiveness and safety of DPJ in 605 patients receiving DPJ or BNJ, 15 days to 6 months after AIS onset. BNJ was used as a control because placebo-controlled studies are not allowed by the Chinese guidelines for TCM clinical research. BNJ had a similar therapeutic function and therapeutic dosage range to DPJ, allowing blinded administration of both products. The patients were assessed after 1 month of treatment by using the Diagnostic Therapeutic Effects of Apoplexy Scale (DTER) with seven items to assess neurological deficits as with the NIH Stroke Scale (NIHSS) and one item to assess functional independence as with mRS [58]. The results of the study showed that DPJ had better functional outcomes than BNJ, as measured by the comprehensive function score component of the DTER. The superiority of DPJ over the control group was statistically significant, and tolerability was excellent in both groups. Using BNJ as a control may have made the interpretation of the results challenging, but since BNJ is an approved TCM for the treatment of AIS in China, it is likely that it has a neutral or positive effect on recovery rather than a negative effect. 

##### Start of the MLC601 Clinical Program

The positive findings of the pooled analysis were encouraging and led to a randomized, double-blind, placebo-controlled trial in patients having experienced an AIS within the past month [88]. Forty patients received either MLC601 or the placebo three times daily for 4 weeks. Fugl–Meyer Assessment (FMA), NIHSS, and Functional Independence Measure (FIM) scores were assessed at baseline and at 4 and 8 weeks. Compared to the placebo, there was a slight favorable trend in improvement observed with MLC601, even up to one month after treatment discontinuation. A subgroup analysis of patients with severe AIS located in the posterior circulation showed a trend toward better recovery with MLC601. This study encouraged larger studies focused on the role of MLC601 in improving recovery from AIS.

##### CHIMES Clinical Program

The Chinese Medicine Neuroaid Efficacy on AIS Recovery (CHIMES) study launched in 2007 was a multicenter, double-blind, placebo-controlled study in which 1100 patients with a moderately severe AIS were randomly assigned to either MLC601 or a placebo for three months [89]. The results for mRS in the entire study population showed an improvement in the MLC601 arm, which was not statistically significant at 3 months. The frequency of serious and non-serious adverse events was similar in both groups. 

Additional analyses showed that age, sex, baseline NIHSS, and onset-to-treatment time (OTT) >48 h were found to be predictors of poorer recovery in the CHIMES Study [90]. Among those patients with at least two predictors of poorer recovery, MLC601 was significantly associated with better outcomes. These subgroup analyses confirm that in order to detect a therapeutic effect in randomized clinical trials (RCTs), it is best to avoid including patients who might have a high chance of spontaneous recovery. 

Following the CHIMES study, the CHIMES-E (Extension) study was conducted to examine the long-term effects of a 3-month course of MLC601 on AIS outcomes up to 24 months after AIS onset [91]. All participants from the CHIMES study were eligible for the CHIMES-E study. They continued to receive standard care and rehabilitation recommended by their healthcare provider. Assessments of the mRS, BI, and significant medical events were performed at 6, 12, 18, and 24 months. A total of 880 patients were included, with blinding of initial treatment allocation maintained. No additional study treatment was provided in the CHIMES-E study beyond the initial 3-month treatment. The results showed that at 6 months, the ORs (95% CI) of favorable outcomes as measured using mRS and the Barthel Index (BI) were 1.49 (1.11; 2.01) and 1.55 (1.14; 2.10), respectively. The favorable odds of functional independence showed statistical significance up to 18 months after AIS. Subsequent analyses in patients with poorer prognosis showed larger treatment effects as was seen earlier in the CHIMES study and with durability of the beneficial effect for up to 2 years after AIS. 

In addition, the treatment effect appeared to be amplified when combined with rehabilitation [92]. The patients from the CHIMES and CHIMES-E databases were stratified by treatment group and whether they had received continued rehabilitation up to month 3. Their functional status was assessed with the mRS and BI at 3, 6, 12, 18, and 24 months. Among the 807 subjects for whom rehabilitation data were available, the mean age was 61.8 ± 11.3 years, and 36% were female. After adjusting for prognostic factors of poor outcome (age, sex, pre-stroke mRS, baseline NIHSS, and time from AIS onset to study treatment), a higher rate of subjects in the MLC601 group showed improvement in functional independence compared to the placebo. This treatment effect in favor of MLC601 was consistently higher among those who received persistent rehabilitation compared to those who did not. This larger effect was sustained at all time points up to 2 years on mRS 0–1 dichotomy, mRS ordinal, and BI ≥95 dichotomy analyses. These findings provide evidence to support the hypothesis that the combination of MLC601 with rehabilitation could yield beneficial and sustained effects on the mechanisms underlying neurorepair after AIS.

An exploratory analysis carried out on patients with an AIS of intermediate severity (NIHSS 8–14 at baseline) demonstrated that those administered MLC601 had a significantly quicker functional recovery (mRS 0–1) compared to those administered the placebo [93]. The MLC601 cohort consistently displayed higher rates of functional recovery compared to the placebo group at all evaluation intervals ranging from 6 to 24 months. Patients in the placebo group took approximately 24 months to achieve a 40% recovery rate, whereas those in the MLC601 group attained the same level of recovery in only 6 months, resulting in a time-saving of around 18 months. This time efficiency not only delivers health benefits but also conserves healthcare resources, particularly for patients with poor prognosis factors such as AIS severity, need for rehabilitation, and delayed treatment access.

##### Vascular Outcomes in the CHIMES Study

Using data from the CHIMES study safety population, post hoc analysis showed that MLC601 reduced early vascular events and deaths, defined as recurrent AIS, acute coronary syndrome, and vascular death, within three months following AIS onset [94]. Compared to the placebo group, the occurrence of early vascular events and deaths was reduced by approximately 50% in the MLC601 group on top of antiplatelet agents and other treatments of cardiovascular risk factors, in the 3 months following AIS onset. This beneficial effect was achieved without an increase in the rates of bleeding and non-vascular deaths. The Kaplan–Meier curves for the composite vascular outcomes showed a noticeable difference in the vascular events and death rates between the two study groups from one month after the AIS (log-rank test P =0.024; hazard ratio 0.51; 95% CI 0.28 to 0.93). In addition to antiplatelet agents, the standard treatment of cardiovascular risk factors included anti-hypertensives, statins, and antidiabetics administered as necessary, which likely contributed to the low rate of vascular events in the CHIMES study. This positive outcome may be related to the reported effect of MLC601 on ATP-dependent potassium (K_ATP_) channels, with a potential role in ischemic preconditioning [95]. 

##### Clinical Studies from Other Countries 

Iran: In a randomized, double-blind clinical trial, the safety and effectiveness of MLC601 (n = 100) were assessed vs. a placebo (n = 50) in Iranian patients after an AIS [96]. The main efficacy endpoint was FMA assessment (see the results in Section Motor functions on page 15). Tolerability was good with only mild and transient adverse events.

Spain: The EPICA study was a multicenter, prospective, observational registry designed to assess neurological and functional recovery over the following 3 months and identify recovery predictors in patients who had experienced a moderate to severe AIS (NIHSS score from 10 to 20) within the preceding 3 months [97]. To be included, patients should have the following data gathered at baseline, one month, and three months: clinical and sociodemographic information, lifestyle advice, rehabilitation prescription, and neurological assessments. Of the 143 patients recruited, 131 (91.6%) were analyzed (mean age 64:9 ± 13:8 years, 49.2% women). Neurorehabilitation of varying intensities was received by 93% of patients. During three months of observation, patients showed improvement in neurological and functional tests. In addition to classical prognosis factors such as age, sex, and AIS severity, an association between the use of MLC901 and better recovery was also detected. Baseline BI score, time to treatment, and MLC901 were identified as predictors of improvement following multivariate analyses. Within the subgroup of patients with more severe AIS (NIHSS > 14), more patients had received MLC901 compared to those who did not and had an improved mRS score above the median at month 1 (71.4% vs. 29.4%; *p* = 0.032) and month 3 (85.7% versus 50%; *p* = 0.058).

Iraq: An observational study assessed the use of MLC601 for 3 months in 217 AIS (200 ischemic and 17 hemorrhagic) patients. MLC601 was given in addition to the patients’ other treatments like antiplatelet, anticoagulant, lipid-lowering, antihypertensive, and hypoglycemic drugs [98]. The patients were assessed medically and neurologically, i.e., motor power, speech, visual field, and mRS, at the time of admission and thereafter monthly for 3 months after discharge from the hospital. Of the 159 ischemic AIS patients with mRS ≥ 2 at baseline, 89 out of 155 (57.4%) at month 1 and 98 out of 134 (73.1%) at month 3 reached mRS 0 or 1. Out of 21 aphasic patients, 9 (75%) showed improvement after 3 months, and 18 out of the 37 patients (48.6%) with visual field defects showed improvement. In total, 20 patients stopped taking MLC601 due to no benefit in 11, a large dose in 5, side effects in 3, and the patient’s wish in 1. The authors concluded that MLC601 is associated with the improvement of post-AIS disabilities.

##### Motor Functions

The effects of MLC601 on motor functions were investigated in two studies. A pilot study in Singapore [88] was followed by a larger study on the effect of MLC601 on post-stroke motor recovery [96]. In this research, 150 patients having suffered an AIS within the past month participated in a double-blind, placebo-controlled randomized clinical trial. Over 3 months, patients were randomly assigned to receive either MLC601 (n = 100) or a placebo (n = 50) on top of standard AIS treatment. Concerning safety, patients in the MLC601 group had few mild and transient adverse events. The authors noted that more patients in the MLC601 group had improved motor recovery as measured using FMA compared to those in the placebo group. Additionally, MLC601 was deemed safe to use on top of standard AIS medications, especially in moderate to severe cases.

##### Cerebral Blood Flow (CBF)

Compared to the placebo the mean change in CBF velocity was significantly greater in the MLC601 group and patients in the MLC601 group also showed a significant improvement in mean mRS and BI scores [99]. The CBF increase with MLC601 in AIS patients might be due to either an effect on the promotion of microcirculation or by the vasodilatory effect of activation of K_ATP_ channels which are important processes contributing to neuroprotection, ischemic preconditioning, and neurorestoration in the CNS [95]. The associated improvement in functional outcomes in the patients supports the neurorecovery effects of the improvement in CBF with MLC601.

##### Visual Defects

A randomized, open-label, controlled trial compared the effects of a 3-month course of MLC601 and piracetam in improving post-stroke homonymous hemianopsia (HH) in 40 patients [100]. At 3 months, visual field defects showed significant improvement compared to baseline (*p* < 0.001), with the relative improvement of right and left visual field defects being 45.7% in the MLC601 group and 30.3% and 32.7% in the piracetam group for the right and left visual fields, respectively. These results indicate that MLC601 may be more effective than piracetam in reducing visual field defects in HH patients, but this did not reach statistical significance due to the small sample size.

##### Systematic Reviews and Meta-Analyses

Systematic reviews and meta-analyses of clinical trials on the use of MLC601 in AIS clinical trials have shown favorable results on functional outcomes (Figure 5) and motor recovery (Figure 6) [89,101]. As all of the studies in the systematic review concerned non-acute strokes with OTT windows of ≤ 1 week to 6 months, the effect of time window was analyzed by including only CHIMES patients who started treatment after 48 h from AIS onset [89]. This poor prognosis factor increased the OR to 1.63 (95% CI, 1.20–2.22, *p* = 0.002) and further reduced heterogeneity, suggesting that the patients treated later in the CHIMES study were more similar to those included in previous studies. 

The overall effects on motor recovery at end of study were also in favor of MLC601 (Figure 6) [101]. 

#### 4.2.2. Clinical Studies on TBI

The clinical development program for MLC901/601 in TBI was initiated approximately 10 years ago in New Zealand and now has four completed studies, three of which have been published.

##### BRAINS Study in New Zealand

The BRAin Injury and Neuroaid Supplementation (BRAINS) study was a randomized, double-blind, placebo-controlled clinical trial, testing the safety and effects of MLC901 on cognitive functioning in 78 patients having experienced within 1 to 12 months a mild to moderate TBI with a Glasgow Coma Scale (GCS) score of 9 to 15 [102]. Secondary outcomes were neurobehavioral sequelae, mood, fatigue, physical disability, and quality of life (QoL). Patients received either MLC901 (n = 36) or a placebo (n = 42) orally at 2 capsules 3 times a day for 6 months, after which they were followed for another 3 months after treatment cessation. 

The MLC901 group demonstrated significant enhancements in complex attention and executive functioning vs. the placebo as assessed on the online neuropsychological test (CNS Vital Signs) at month 6, with group allocation being independently predictive of complex attention. At month 6, there was a mean difference of 11.88 for complex attention and 7.16 for executive functioning, implying clinically meaningful improvements with moderate to small effect sizes (d = 0.6 and d = 0.4 respectively). The MLC901 group exhibited cumulative gains in their complex attention trajectories over time, whereas the control group remained relatively stable. Executive functioning improved in both groups, with the MLC901 group improving more quickly. Following the endpoint "end of treatment", there was a decline in cognitive function improvements from month 6 to month 9. Throughout the study duration, both groups exhibited enhancements in all secondary outcomes. Despite several positive trends favoring MLC901, there were no significant differences between the groups in the neurobehavioral consequences of mood, tiredness, and general QoL at the *p* < 0.05 level across different time intervals. Individuals in the MLC901 group displayed significant gains in the cognitive aspect of QoL, with a mean difference of 1.62 (Cohen’s d = 0.1).

##### NEPTUNE Study in Indonesia

In the NEPTUNE (NEurological Prognosis after brain Trauma and Use of NeuroAiD) study, the effects of MLC601 administered within 2 days of the traumatic event on functional and neurological outcomes in patients with moderate non-surgical TBI were evaluated (GCS 9 to 13) [103]. In this study, 32 patients were randomized to receive either MLC601 capsules three times a day for 3 months (n = 16) or no MLC601 (n = 16) serving as a control group, with a 6-month follow-up. 

Upon discharge, the median Glasgow Outcome Scale (GOS) in the MLC601 group slightly exceeded that of the control group, registering at 3 and 2.5, respectively, although this difference did not reach statistical significance. Both groups displayed a significant enhancement in GOS scores from the point of discharge to month 6. The analysis of GOS trajectories over time showed that they were higher in the MLC601 group at month 6 compared to the control group, but this was not statistically significant. In contrast, BI assessment demonstrated higher median values in the MLC601 group compared to the control group at all time points, reaching significance at month 3 (47.5 vs. 35.0; *p* = 0.014) and month 6 (67.5 vs. 57.5; *p* = 0.055). The trajectories of BI scores over time showed significant improvement of BI in the MLC601 group from time of discharge to month 3 and to month 6 (40.0 to 47.5 to 67.5, respectively).

##### Pilot Study in Moderate to Severe TBI in Iran

The primary objective of this placebo-controlled study was to evaluate the efficacy of MLC901 in improving neurological outcomes among individuals with moderate to severe TBI [104]. A total of 98 patients were included within 24 h of injury, with a non-penetrating moderate or severe TBI. They were randomly allocated to receive either MLC901 or a placebo, administered as 2 capsules thrice daily for 6 months. The effectiveness endpoints were assessed with GOS and mRS. 

A total of 81 patients successfully completed the 6-month follow-up period, with the assessment of GOS and mRS scores at each follow-up session. At study entry, demographic characteristics, OTT, and length of stay in the intensive care unit were comparable between groups. The results showed that individuals receiving MLC901 had significantly improved functional outcomes, as demonstrated by elevated GOS scores, at 3- and 6-month post-injury assessments (*p* = 0.023 and *p* = 0.006, respectively). Similarly, the mRS scores were significantly lower in the MLC901 group at 3 months (*p* = 0.024) and 6 months (*p* = 0.04) compared to the placebo group. 

These results indicate that the prompt administration of MLC901 could lead to improved disability and functional recovery in patients with moderate to severe TBI. The improvements obtained in both assessment scales over the 6-month period provide further evidence supporting the favorable effects of MLC901 as an early therapeutic approach for this patient group.

##### SAMURAI Study in Russia

Preliminary results of the SAMURAI placebo-controlled study designed to assess the benefit and safety of a 6-month treatment with MLC901 in patients with mild TBI were presented at WCN 2023 in Montreal [105]. A total of 182 participants (MLC901, n = 92; placebo, n = 90) were enrolled and randomized into the study. 

As opposed to the BRAINS study [102], this study did not detect a statistically significant difference with a placebo for the cognitive patient-reported computerized assessment of complex attention (primary endpoint). However, the study showed a clinically and statistically significant improvement in all clinical scales assessed by the investigators on the Rivermead Post-concussion Symptoms Questionnaire (RPQ), Quality Of LIfe after BRain Injury (QOLIBRI), and the Hospital Anxiety and Depression Scale (HADS).

## 5. Discussion

Although they do not share similar mechanisms at the acute stage, AIS and TBI are more related in the post-acute phase, where the self-repair processes of the brain are activated. If the adage “Time is the brain” applies well to the acute phase, then “The brain needs time to recover” might be more appropriate for the next phase of rehabilitation. However, for a majority of subjects, natural neurorepair processes are often insufficient to achieve complete recovery of their impaired functions.

Therefore, one of the fundamental pillars of post-stroke care continues to be neurorehabilitation. Two studies showed that the addition of MLC601 to rehabilitation resulted in enhanced functional recovery and independence in patients, surpassing the outcomes of those undergoing rehabilitation alone [91]. The therapeutic effects were observed as early as three months during MLC601 administration and continued to increase over time, reaching their peak at one year. This indicates a potential positive influence on the brain’s healing mechanisms post AIS. Another analysis provided an important benefit of MLC601, showing that compared to a placebo, the time to achieve functional recovery is shortened by 18 months [93]. It may be assumed that these cumulative benefits could have a favorable impact on the quality of life of patients.

AIS and TBI have other connections, such as those related to reciprocal risk factors. Indeed, systematic reviews and meta-analyses have demonstrated a statistically significant correlation between head trauma and the long-term probability of suffering an AIS [106,107,108]. The risk of AIS, and particularly hemorrhagic stroke, has been shown to be higher in the first year after injury, especially in people with more severe head injuries. Therefore, it is crucial to emphasize the importance of diligently addressing vascular risk factors and implementing primary AIS prevention strategies in individuals with previous head trauma. In order to improve methods of secondary prevention of AIS, reduce morbidity and mortality related to head injuries, and better understand the processes causing the high risk of AIS, it may be useful to fully understand this link. 

Likewise, the correlation between AIS and susceptibility to head trauma is well exemplified. An observational study of individuals who experienced a minor AIS showed that these individuals were more than twice as likely to experience falls compared to healthy controls [109]. This study highlights the potential repercussions of a minor AIS on aspects such as balance and gait capacity, confidence in balance, risk of falling, and daily physical activities. Deficiencies in balance and walking capabilities were evident across all clinical assessments. A noteworthy finding from this study is that even individuals who had fully recovered displayed inferior balance and walking capacities compared to healthy controls. These results suggest that balance and walking ability after a minor AIS could be influenced by clinically imperceptible leg motor impairments or higher-level sensorimotor coordination, both of which pose a substantial risk for falls [109]. With twice more falls in this mild AIS cohort, the balance and gait impairments are clinically relevant. Individuals having experienced a mild AIS should be considered for training to improve balance and/or gait capacity, even in the absence of motor function deficits.

MLC601 and MLC901 are already in clinical use. Collectively, the current efficacy data have recently allowed these formulations to be included in the recommendations of two international guidelines. For stroke, a group of experts from the Asian Stroke Advisory Panel gave them a B-R (randomized) level recommendation in 2024, defined as moderate quality evidence from one or more RCTs and meta-analyses of moderate quality [110]. For TBI, in the International Cognitive Sciences Group of Researchers and Clinicians (INCOG 2.0) guidelines, MLC901 is recommended with the highest level (A) of evidence for improving complex attention in people with mild to moderate TBI [111]. On the other hand, all of the pharmacoclinical and clinical results achieved with MLC901 support the development of a new 4-ingredient formulation, MLC1501, which has obtained US FDA IND for both stroke and TBI indications. A clinical trial is ongoing as an add-on treatment for motor and post-stroke recovery (NCT05046106).

Before closing this review, we should say a word about a more or less long-term consequence of AIS and TBI, namely the occurrence of Alzheimer’s disease or other forms of dementia. MLC901/601 has demonstrated clinical benefits in both early treatment compared to standard symptomatic treatments (SSTs) [112] and in combination with SSTs [113]. The main results of this development are presented in a recent review paper [112]. It is imperative to find effective treatments that can either prevent or delay the progression of the disease in order to obtain accurate diagnostic tools for confirming the diagnosis of AD and monitoring its long-term therapy. There is increasing agreement on the importance of adopting a multi-factorial approach to treatment and developing appropriate combinations of drugs for AD. Traditional medicines have long been advocating for this approach. MLC901, for instance, can be safely integrated either after unsuccessful symptomatic treatments or in addition to them [112]. 

## 6. Conclusions

At the end of this review of AIS and head injuries, certain lessons are worth reiterating. First, the burden linked to these two pathologies remains heavy and is expected to worsen in the next two decades. Secondly, although their appearance is linked to distinct causes, a large part of the neurobiological disorders they cause are similar, once their acute phase has passed. Thirdly, with these disorders being numerous and complex, it is necessary to use a multimodal and safe therapeutic approach, as is the case with TCMs. Fourth, through its preclinical and clinical development results, MLC901 meets most of these conditions. This supports the beneficial effect of MLC901 on recovery for individuals with remaining impairments after a stroke or TBI.

## Figures and Tables

**Figure 1 biomolecules-14-00680-f001:**
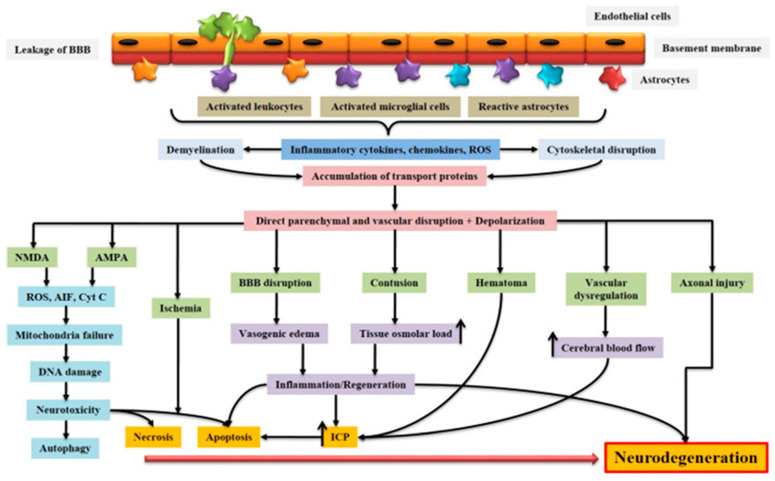
Schematic representation of the pathophysiology of acquired brain injury. BBB dysfunction caused by injury allows for the transmigration of activated leukocytes into the injured brain parenchyma, which is facilitated by the upregulation of cell adhesion molecules. Activated leukocytes, microglia, and astrocytes produce ROS and inflammatory molecules such as cytokines and chemokines that contribute to demyelination and disruption of the axonal cytoskeleton, leading to axonal swelling and the accumulation of transport proteins at the terminals. On the other hand, excessive accumulation of glutamate and aspartate neurotransmitters in the synaptic space due to spillage from severed neurons activates NMDA and AMPA receptors located on post-synaptic membranes, which allow for the production of ROS. As a result of mitochondrial dysfunction, molecules such as apoptosis-inducing factor (AIF) and cytochrome c are released into the cytosol. These cellular and molecular events including the interaction of Fas with its ligand Fas ligand (FasL) ultimately lead to caspase-dependent and -independent neuronal cell death. BBB: blood–brain barrier; NMDA: N-methyl-D-aspartate receptor; AMPA: α-amino-3-hydroxy-5-methyl-4-isoxazole propionic acid receptor; ROS: reactive oxygen species; Cyt c: cytochrome c; ICP: intracranial pressure; AIF: apoptosis-inducing factor [10]. Creative Commons Attribution-Available via license: CC BY 4.0 [10].

**Figure 2 biomolecules-14-00680-f002:**
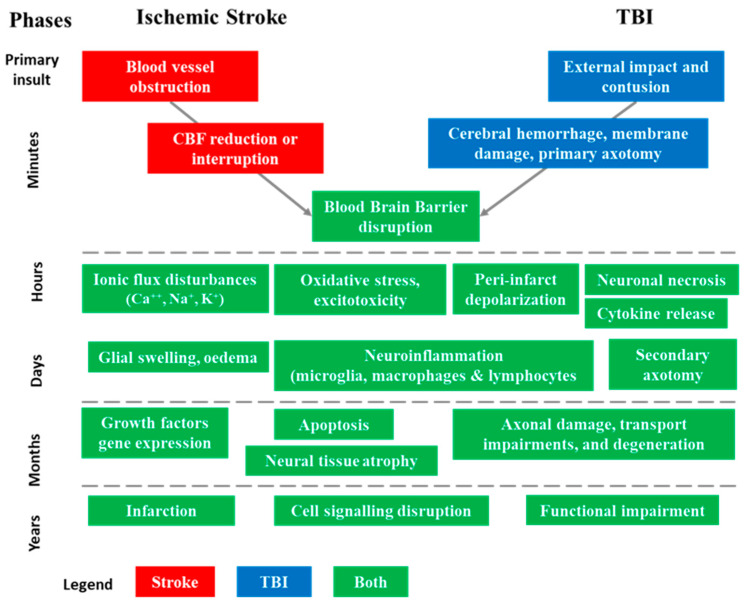
Sequence of pathophysiological events involved in acute ischemic stroke (AIS) and traumatic brain injuries (TBI) in their successive phases [9,17]. The injury process can be divided into four distinct phases: primary insult (minutes), acute (hours), subacute (days–weeks), and chronic outcome (months–years). These phases help in understanding the progression of structural and functional abnormalities associated with AIS and TBI.

**Figure 3 biomolecules-14-00680-f003:**
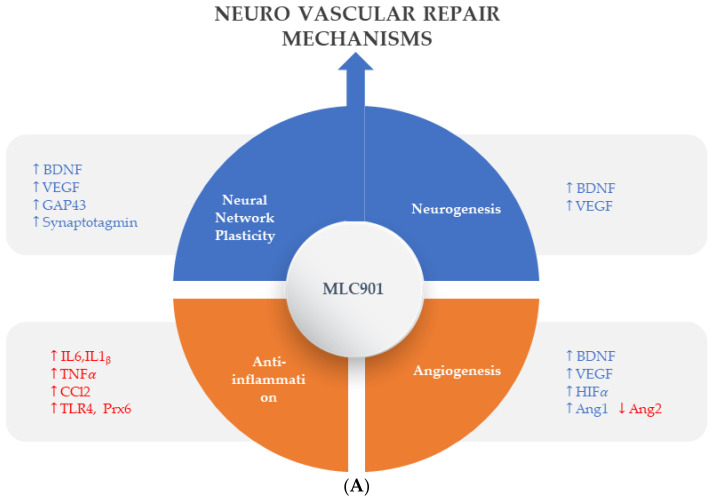
Regulated effects of MLC901 on key neurovascular repair mechanisms in A—AIS and B—brain trauma. (**A**) Preclinical models: MLC901 enhances these neurovascular repair mechanisms with its multimodal actions of up- (blue arrows) or down (red arrows)-regulation on growth factors (BDNF, VEGF, and angiopoietins 1 and 2), cytokines (TNFα and interleukins), chemokines (CCL2), and other mediators (HIFα, TLR-4, and Prx-6). MLC901: red arrows for downregulation; blue arrows for upregulation. BDNF: brain-derived neurotrophic factor; VEGF: vascular endothelial growth factor; CCL2: chemokine (C-C motif) ligand 2; HIFα: hypoxia-inducible factor 1-alpha; TLR4: Toll-like receptor 4; Prx6: peroxyredoxin-6. (**B**) Pathological events happening during primary and secondary phases of traumatic brain injury (TBI) with a description of the short-term and long-term consequences of brain trauma. The blue font shows the phases of TBI that MLC901 can regulate. ↑ shows the increased oxidative stress, ↓ shows reduced oxidative stress [59]. Creative Commons Attribution-Available via license: CC BY 4.0–Ref. [59].

**Figure 4 biomolecules-14-00680-f004:**
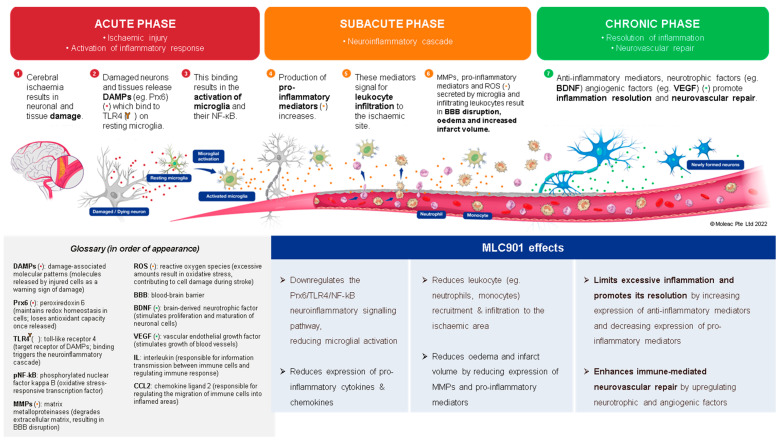
Effects of MLC901 on the post-stroke neuroinflammatory cascade and immune-mediated neurovascular repair. © Moleac Pte. Ltd., Singapore.

**Figure 5 biomolecules-14-00680-f005:**
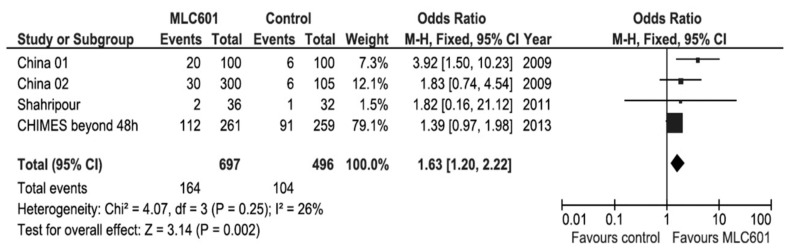
Meta-analysis of functional outcomes with data of patients treated beyond 48 h after stroke in the CHIMES study [89].

**Figure 6 biomolecules-14-00680-f006:**
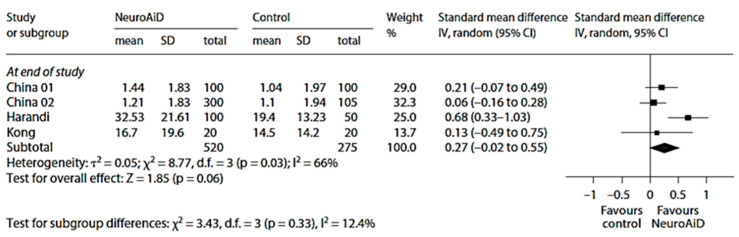
Meta-analysis of motor outcomes at the end of study from four studies on MLC601. Adapted from [101]. The final, published version of this article from which the figure has been adapted is available at https://www.karger.com/?doi=10.1159/000346231, accessed on 8 March 2024.

**Table 1 biomolecules-14-00680-t001:** Multimodal mode of action of MLC901, a natural multi-herbal formulation [57].

MLC901’s Multimodal Mode of Action	MLC901 Ingredients
Effects on brain lesion recovery (1): brain tissue	
- Promotes neuroplasticity and prevents the loss of axons and synapses	*Radix Astragali*
- Promotes neurogenesis and slows down neurodegeneration	*Radix Astragali and Rhizoma Acori tatarinowii*
- Anti-inflammatory effect	*Radix Salviae miltiorrhizae and Carthamus tinctorius Semen persica*
- Improve hypoxia	*Radix Astragali, Radix Salviae miltiorrhizae, Radix Paeonia rubra, and Carthamus tinctorius*
Effects on brain lesion recovery (2): cerebral vascularization	
- Increases cerebral blood flow	*Rhizoma Chuanxiong and Semen persica*
- Improve microcirculation	*Rhizoma Chuanxiong, Radix Salviae miltiorrhizae, and Carthamus tinctorius*
- Protects the cardiovascular system	*Radix Astragali*
- Antithrombotic effect	*Radix Astragali, Rhizoma Chuanxiong, Radix Angelicae sinensis, Radix Paeonia rubra,* *and Semen persica*
- Anti-atherosclerotic properties	*Radix Salviae miltiorrhizae* *and Carthamus tinctorius*
- Improves hemodynamics, reduces vascular resistance, and reduces blood viscosity	*Radix Salviae miltiorrhizae Rhizoma, Chuanxiong, Carthamus tinctorius, and Semen persica*
Effects on cognition and memory	
- Improves cognitive dysfunction	*Radix Polygalae and Rhizoma Acori tatarinowii*
- Attenuates memory deficits	*Radix Astragali, Radix Polygalae, Carthamus tinctorius, Semen persica, and Rhizoma Acori tatarinowii*
Cardiac protection	
- Protective effect on myocardial ischemia and increase in myocardial blood flow and oxygen supply	*Rhizoma Chuanxiong, Radix Angelicae sinensis* *Radix Salviae miltiorrhizae, and Carthamus tinctorius*
- Antagonizes arrhythmia	*Radix Astragali, Radix Salviae miltiorrhizae* *Carthamus tinctorius, and Rhizoma Acori tatarinowii*
Protection against CV risk factors	
- Antihypertensive	*Rhizoma Chuanxiong and Radix Salviae miltiorrhizae*
- Lipid lowering	*Radix Astragali, Radix Salviae miltiorrhizae,* *and Carthamus tinctorius*
Effects on frequent diseases concomitant to brain lesions	
- Anti-infectious	*Radix Salviae miltiorrhizae, Radix Paeonia rubra, and Rhizoma Acori tatarinowii*
- Anticonvulsant	*Radix Paeonia rubra, Carthamus tinctorius, and Rhizoma Acori tatarinowii*

## Data Availability

All data generated or analyzed during this review are included in the article, along with references to data from cited published studies. Further inquiries can be directed toward the corresponding author, Narayanaswamy Venketasubramanian.

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
