# Peer review of "Translational Medicine in Acute Ischemic Stroke and Traumatic Brain Injury—NeuroAiD Trials, from Traditional Beliefs to Evidence-Based Therapy"

_biomolecules, 2024, doi:10.3390/biom14060680_

Round 1
Reviewer 1 Report
Comments and Suggestions for Authors
In this manuscript Venketasubramanianand colleagues present an article reviews similarities and differences between stroke and traumatic brain injury in their consequences and pathophysiological processes. It focuses on a post-acute intervention, NeuroAiD, a natural product with a multitarget mode of action, derived from traditional medicine and supported by evidence. This treatment has shown promising results in preclinical and clinical studies for both conditions.
The manuscript provides a comprehensive and accurate review of the epidemiology, pathophysiology, and therapeutic approaches related to cerebrovascular accident and traumatic brain injury. The images included are informative, although some could be simplified to improve understanding or divided into smaller panels. An extensive portion of the paper is dedicated to the development of MLC901, explaining its mechanisms of action and showing its regulated effects on key neurovascular repair mechanisms in stroke and TBI. Additionally, significant clinical trial meta-analysis results on the use of MLC601 in stroke clinical trials are presented, further reinforcing the therapeutic relevance of the study. Overall, the manuscript offers a valuable contribution to the field of clinical neuroscience and neuroprotective therapy.
The therapeutic approaches related to stroke and traumatic brain injury require a deeper, more extensive, and critical analysis. For example, the role of non-excitat​ory amino acids and their transporters as potential therapeutic targets should be discussed. Discuss their strengths, limitations, and implications.
Comments on the Quality of English Language
The manuscript is verbose and requires additional editing of English.
Author Response
Dear Reviewer, thank you very much for your support and helpful suggestion. We have revised accordingly:
- The therapeutic approaches related to stroke and traumatic brain injury require a deeper, more extensive, and critical analysis. For example, the role of non-excitat​ory amino acids and their transporters as potential therapeutic targets should be discussed. Discuss their strengths, limitations, and implications.
Authors’ reply: we have now added a new paragraph about non-excitatory amino-acids (lines 195-202)
‘Another pathway to consider is that of non-excitatory amino acids (L-alanine, glycine, L-histidine, L-glutamine, L-serine, L-threonine, and taurine) which trigger cellular swelling under hypoxic conditions, by increasing levels of extracellular glutamate, the principal excitatory neurotransmitter [21]. Thus, these non-excitatory amino acids (AA) could promote harmful effects by activating N-methyl-D-aspartate (NMDA) receptors (NMDAR). However, regarding the inefficacy of NMDAR blockade observed in clinical trials with AIS patients, an alternative to reduce ischemic damage might be targeting the AA transporters involved in this process’.
- Comments on the Quality of English Language
The manuscript is verbose and requires additional editing of English.
Authors’ reply: we sincerely apologise for this - we have tried to reduce the verbosity and edited the English best we can.
We hope this is adequate.
Thank you again
Reviewer 2 Report
Comments and Suggestions for Authors
The review paper explores the significance and pathological development of ischemic stroke and traumatic brain injury. Consequently, the review paper summarises the preclinical and clinical evidence for the use of MLC901 (NeuroAiDTMII), a natural multi-herbal formulation, for an acute treatment. The review is well written which address a global and significant problem. I have a few comments for minor revision as shown below
1. Please consider adding “ischemic” to the review title
2. Would it be possible to reveal the dosage of the ingredients in the formulation of MLC901 (NeuroAiDTMII)? Or at least let reader know which are the main/king herbs. Is this a traditional formulation or derived from a classic formulation? Is it already used in clinical practice?
3. If the information in Table 1 is based on studies and literature, please include references; please also consider to re-structure the table to improve the clarity or even change it into a figure/diagram
4. Please consider improving the quality of figure 3, it doesn’t look professional…
5. It is very rare to see a figure is cited in a subheading – 4.1.1, 4.1.2. to 4.1.3
6. Words become very small in Figure 4, please improve. Also is it derived from a previous study, if so, please add reference
7. Question, what is the main barrier to introduce the product globally?
Author Response
Dear Reviewer, thank you very much for your many helpful suggestions. We have revised accordingly:
- Please consider adding “ischemic” to the review title
Authors’ reply: Thank you for the suggestion. Adding has been done of “ischemic” in the title (line 2).
- Would it be possible to reveal the dosage of the ingredients in the formulation of MLC901 (NeuroAiDTMII)? Or at least let reader know which are the main/king herbs. Is this a traditional formulation or derived from a classic formulation? Is it already used in clinical practice?
Authors’ reply: Thank you for this important question. This is derived from a traditional formulation with only herbal components – these have now been clarified and and listed (lines 377-381). Yes, both MLC601 and MLC901 are in clinical use - this information has now been added (line 961).
- If the information in Table 1 is based on studies and literature, please include references; please also consider to re-structure the table to improve the clarity or even change it into a figure/diagram
Authors’ reply: Thank you for the suggestion. Table 1 has been set up based on information from ref 57 (Heurteaux et al. 2013) - this ref is now added in the Table title.
- Please consider improving the quality of figure 3, it doesn’t look professional…
Authors’ reply: We apologise for this. The Figure 3 has now been revised.
- It is very rare to see a figure is cited in a subheading – 4.1.1, 4.1.2. to 4.1.3
Authors’ reply: We apologise, we have now removed the inappropriate subheading.
- Words become very small in Figure 4, please improve. Also is it derived from a previous study, if so, please add reference
Authors’ reply: We again apologise. Figure 4 has now been revised with improved readability. It is a Figure made in-house by Moleac who provided it. We have added in the title the Moleac copyright which was in the image.
- Question, what is the main barrier to introduce the product globally?
Authors’ reply: This is an important question. But with due respect, as we are practicing doctors independent of Moleac which owns the product, we are unable to answer this marketing, our sincere apologies.
We hope this is adequate.
Thank you again